# Intraoperative Control of Hemoglobin Oxygen Saturation in the Intestinal Wall during Anastomosis Surgery

**Daniil M. Kustov** [1,*] **, Tatiana A. Savelieva** [1,2] **, Timofey A. Mironov** [2] **, Sergey S. Kharnas** [3] **, Vladimir V. Levkin** [3] **, Andrey S. Gorbunov** [3] **, Artem A. Shiryaev** [3] **and Victor B. Loschenov** [1,2]

1 Prokhorov General Physics Institute of the Russian Academy of Sciences, 119991 Moscow, Russia; savelevat@nsc.gpi.ru (T.A.S.); loschenov@nsc.gpi.ru (V.B.L.)
2 Institute of Engineering Physics for Biomedicine, National Research Nuclear University MEPhI, 115409 Moscow, Russia; mironov.timofei@mail.ru
3 Department of Faculty Surgery No. 1, I.M. Sechenov First Moscow State Medical University, 119992 Moscow, Russia; kharnas_s_s@staff.sechenov.ru (S.S.K.); levkin_v_v@staff.sechenov.ru (V.V.L.); gorbunov_a_s@staff.sechenov.ru (A.S.G.); shiryaev_a_a@staff.sechenov.ru (A.A.S.)
* Correspondence: kustovdm@bk.ru or kustov.dm@nsc.gpi.ru

**Abstract:** During surgery for colon cancer, monitoring of the oxygen saturation of hemoglobin in the tissues under study makes it possible to assess the degree of blood supply to the anastomosis areas of the colon. Adequate blood supply in this area is decisive in terms of the consistency of the anastomosis and can significantly reduce the risk leakage of anastomosis. In this work, we propose a new approach to assessing the hemoglobin oxygen saturation based on measuring both the diffuse reflectance and transmittance spectra of the colon wall tissues. The proposed method is based on the use of two fiber-optic tools for irradiation from both sides—the intestinal lumen and the outside of the intestinal wall. The spectra are recorded from the external side. To determine the degree of hemoglobin saturation, two algorithms, both based on the Taylor series expansion of the coefficient of light attenuation by tissues, are proposed. The results of a clinical study of the proposed approach on volunteers were obtained, allowing to draw a conclusion about the applicability of the approach in a clinical setting.

**Keywords:** optical spectroscopy; diffuse transmittance; diffuse reflectance; oxygen saturation of hemoglobin; intestine tissues

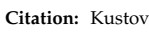



## 1. Introduction

Oxygen saturation of hemoglobin in the colon wall and the tumor volume is an important diagnostic parameter and prognostic factor [1]. One of the important applications of this diagnostic parameter is surgery to remove colon cancer. During operation, an adequate blood supply in the anastomotic area is decisive to assessing its viability and can significantly reduce the risk of developing peritonitis [2]. Oxygenation, i.e., the binding of heme to oxygen, leads to a transition of iron from a high to a low spin state, a change in the bond length, and the retraction of an iron atom into the plane of the heme ring. These conformational changes affect the electronic absorption spectrum, which makes it possible to determine this crucial diagnostic parameter by optical methods.

Determining the level of oxyhemoglobin in tissues is of interest in medicine. Creating a surgical anastomosis after resectioning the colon tumor is usually performed without complications [3]. If the healing of the anastomosis is unsuccessful, the contents of the intestinal lumen can leak into the abdominal cavity and cause septic peritonitis, leading in the worst case to a fatal complication. Generally, the anastomotic leak rate is 5–7% but can reach 24% for operations in the rectum [4,5]. In surgical operations, intraoperative analysis of hemoglobin oxygen saturation will make it possible to assess the state of the blood supply to the organs to be sutured for the prevention of anastomotic suture failure and graft necrosis.

One of the obvious choices for risk of anastomotic leak assessment is an intraoperative fluorescence angiography [6]. Near-infrared indocyanine green (ICG) fluorescence guiding for detecting microvascular impairment is a cutting-edge technology and potentially can prevent anastomotic leakage [7,8]. However, it doesn't help to analyze hemoglobin oxygen saturation only the structural integrity problems.

Methods such as laser Doppler and white light spectroscopy are used to study the blood oxygen saturation in the intestinal wall [9]. For white light spectroscopy, the authors used the flat probe containing two light sources and two detectors for measuring the scattered light. It worked with a white light source in a wavelength range of 500–630 nm and a laser light source at 830 nm. The penetration depth of both was approximately 2.5 mm.

The research of the possibility of leak prediction with intraoperative colonic pulse oximetry during colorectal surgery is presented in the article [10]. The probe has a light source that emits red and infrared light (660 and 910 nm) and a light sensor. The arterial hemoglobin oxygen saturation level is calculated from the pulsative part of the light signal by analyzing the amount of light absorbed by hemoglobin.

In work [11], the predictive value of visible light spectroscopy (VLS) to measure tissue oxygenation for anastomotic leakage of the colon and the rectum was evaluated. The authors used a spectral oximeter with a sensor containing a white LED light to illuminate the tissue in the capillary bed. Then the signal is returned through a fiber optic bundle for real-time spectroscopic analysis.

The main purpose of this study is to analyze the possibility of intraoperative assessment of the degree of hemoglobin oxygen saturation in the intestinal wall tissues both in diffusely reflected light and in transmitted radiation. To take into account the effect of scattering, other tissue absorbers and non-uniform distribution of hemoglobin on the received signal, the difference spectra were modeled based on the experimental data obtained in the position of the study on the one side (OS) and "light through" (LT). We used the approach obtained on the basis of a combination of the diffusion approximation model with the expansion of the coefficient of light attenuation by tissues in the Taylor series [12] and the empirical model proposed by S. Jacques [13]. In this work, we propose an approach to measure oxygenation not only in the geometry of back-reflected radiation, as in the various previous works described above, but also in transmitted light. As it will be shown below, measurements in transmitted light are more stable and, therefore, may serve as a more reliable prognostic factor.

## 2. Materials and Methods

### 2.1. Spectroscopic Setup

The experimental setup (Figure 1a) consisted of the white light source, spectrometer with fiber optic input, fiber optic probe, PC with special software for spectrum registration and processing. For spectrum registration, the spectrum analyzer LESA-01-BIOSPEC was used. In the 500–600 nm spectrum range, the recording of diffuse reflection and transmission spectra was implemented, which allows for a quantitative assessment of the concentration of hemoglobin in oxygenated and reduced form.

A light-emitting diode lamp with a fiber-optic output was used as a source of broadband light. The power of the light source was regulated within the limits of up to 1 W, and the color temperature was 5000 K–6500 K. The spectrum of the light source in the range of 500–600 nm had a uniformity sufficient for the analysis of diffuse scattering spectra. Diffuse reflectance spectra were recorded relative to a standard sample ($BaSO_4$) with a reflectance close to unity in the spectral range of interest to us (500 nm–600 nm).

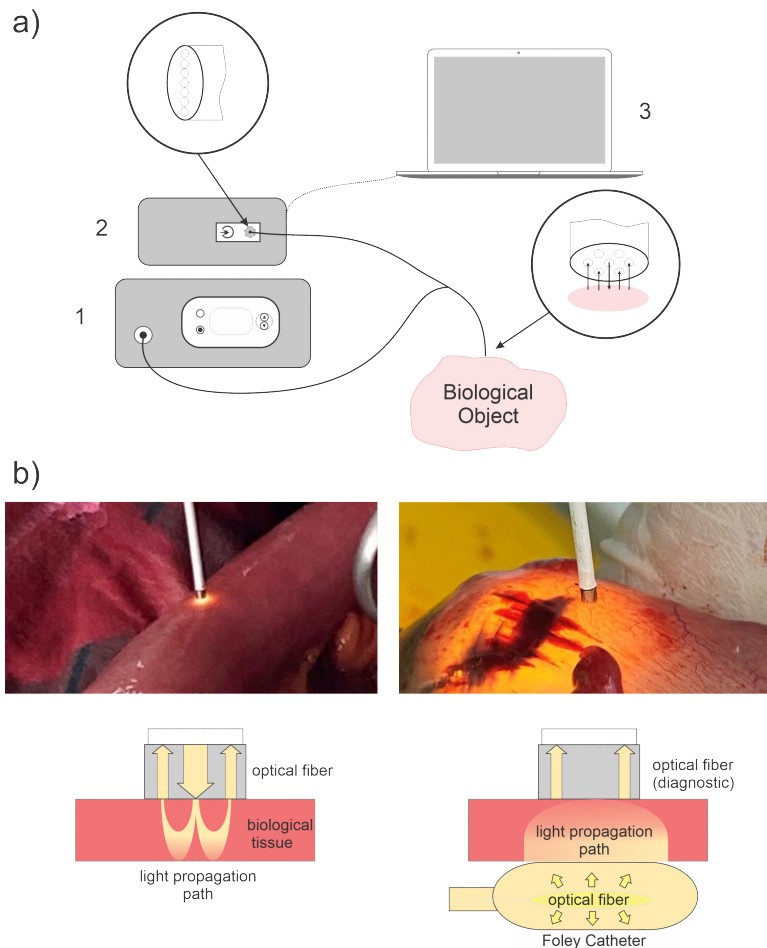

**Figure 1.** (**a**) The experimental setup: white light source (1), spectrometer (2) with fiber optic input, Y-shaped fiber optic probe, and PC with special software (3) for spectrum registration and processing. The fiber optic probe consisted of 7 fibers: 6 for receiving and 1 for transmitting light radiation. The location of the receiving fibers around the emitting one allows reducing the influence of the inclination of the end of the bundle on the received signal. (**b**) The geometry of diffuse scattering spectra measurements: in reflected light (**left**) and in transmitted light (**right**).

When placing fibers on both sides of the intestinal wall for measurement in transmitted light, a fiber optic polymer diffuser was placed inside the Foley catheter. That was used to uniformly illuminate the intestinal wall from the inside in order to measure diffusely scattered light that passed through the intestinal wall to the outside (Figure 1b, right). The Foley catheter was filled with a fat emulsion (Intralipid MCT/LST 10%) diluted to 1% NaCl 0.9%, selected as a scattering medium to ensure a uniform scattering indicatrix. The length of the diffuser was 10 mm.

A standard fiber optic probe (included in LESA-01-BIOSPEC) with a central lighting fiber and six peripheral receiving fibers was used both for illumination (in the case of placing the fibers on one side) and for recording: diffusely reflected (Figure 1b, left) and diffusely transmitted light (Figure 1b, right). During the measurement, the end face of the fiber optic probe was in soft contact (without fiber pressure on the tissue) with the biological tissue.

The receiving fibers at the entrance to the spectrometer form a line serving as the entrance slit of the monochromator and were positioned perpendicular to the diffraction plane of the dispersing element. The diameter of each cladded fiber was 250 μm, which determined the distance between the centers of the fibers located close to each other. The numerical aperture of each fiber is 0.22.

### 2.2. Algorithm for Assessing the Hemoglobin Oxygen Saturation

To assess the saturation of hemoglobin with oxygen from the diffuse scattering spectra, an algorithm is proposed that separates the recorded spectrum into components due to absorption and scattering [14]:

$$A = ln(\frac{I_0}{I}) = c_0 + c_1\lambda + c_2\lambda^2 + \langle L \rangle \cdot \{c_{Hb} \cdot \varepsilon_{Hb}(\lambda) + c_{HbO_2} \cdot \varepsilon_{HbO_2}(\lambda)\} \cdot ln(10), \quad (1)$$

where $A$ is the light attenuation coefficient, $I_0$ is the intensity of incident light (measured during system calibration as reflected light from a reflective standard ($BaSO_4$), $I$ is registered light (in diffusevely reflected and transmitted light), $\lambda$ is the wavelength (nm), $\langle L \rangle$ is the mean pathlength of photons in tissue between source and receiving fibers, $c_{Hb}$ is the reduced hemoglobin concentration, $c_{HbO_2}$ is the oxygenated hemoglobin concentration, $\varepsilon_{Hb}$ is the molar extinction coefficient of the reduced hemoglobin, $\varepsilon_{HbO_2}$ the molar extinction coefficient of oxygenated hemoglobin, and $c_i$ is fitting coefficients. The value of each $c_i$ as well as the values of $c_{Hb}$ and $c_{HbO_2}$ are obtained by minimizing the objective function:

$$\chi^2 = \sum_{\lambda_i=\lambda_{min}}^{\lambda_i=\lambda_{max}} [A_{model}(\lambda_i) - A_{exp}(\lambda_i)]^2, \quad (2)$$

where $\lambda$ is the wavelength, limited by the spectral range in which we approximate the experimental spectrum with the proposed model. The equation aims to minimize the chi-square of the sum of residuals of fitting the model data to the experimental data.

Obviously, this approach makes it possible to calculate the concentrations of oxy- and deoxyhemoglobin only up to the value of the average path of photons in the tissue, which can be approximately determined through the distance between the lighting and receiving fibers. However, when calculating the oxygen saturation of hemoglobin, the factor corresponding to the length of the light path in the tissue is reduced since the degree of saturation was calculated as the ratio of the concentration of oxyhemoglobin to the total concentration of oxy- and deoxyhemoglobin:

$$SO_2 = \frac{c_{HbO_2}}{c_{HbO_2} + c_{Hb}}, \quad (3)$$

where $c_{Hb}$ is the reduced hemoglobin concentration, and $c_{HbO_2}$ is the oxygenated hemoglobin concentration. In the model, the reduced scattering coefficient was calculated using the power function:

$$\mu's(\lambda) = a_0(\lambda/\lambda_0)^{-a_1}, \quad (4)$$

where $\lambda_0$ = 550 nm.

In order to estimate the absolute value of the concentration of hemoglobin in oxygenated and reduced form, it is possible to use data on the average pathlength of photons in tissues between the emitter and the receiver based on other models. For this purpose, we used a numerical simulation of the light propagation in tissues by the Monte Carlo method with the calculation of a photon path length distribution function. For a numerical aperture of 0.22 and a distance between the emitting and receiving fibers of 250 µm, the mode of such a distribution function fell on 570 ± 12 µm.

The optical properties of individual layers of the intestinal wall as input data for numerical modeling were taken from publications [15–18]. As a result of analysis of data from these sources, the absorption coefficient was taken equal to 6.5 cm$^{-1}$ for the muscle layer, 1 cm$^{-1}$ for the submucosa, and 3.5 cm$^{-1}$ for the mucosa. The scattering coefficient and the anisotropy factor were taken equal to 100 cm$^{-1}$ and 0.89 for the muscle layer, 100 cm$^{-1}$ and 0.9 for the submucosa, and 300 cm$^{-1}$ and 0.9 for the mucosa, respectively.

### 2.3. Biological Object of Research

The study was carried out based on the Faculty Surgery Clinic №1 named after N.N. Burdenko of I.M. Sechenov First Moscow State Medical University. It was conducted on 10 volunteers who underwent resectioning of a tumor of the sigmoid colon with anastomosis and optical-spectral monitoring of tissue oxygen saturation ($StO_2$) in the colon, including the anastomotic region. For them, the data are presented averaged over the sample.

As a demonstration of the capabilities of the method for determining the level of oxyhemoglobin, we present a specific clinical case in this article too; patient P., age 49, with tumor of the sigmoid colon T4aN0M0 (T4a—the tumor grows through all layers of the intestinal wall and does not grow into the surrounding tissues; N0—no metastases in lymph nodes; M0—no distant metastases), moderately differentiated adenocarcinoma (grade 2). The tumor size was 6 cm × 5 cm × 5 cm.

Low-intensity radiation with a power of 1–5 mW was used to avoid light thermal and destructive effects on the intestinal wall tissues. The diagnostic catheter was in soft contact (without pressure on the tissue) with the intestinal wall when recording the spectra. The fibers were placed perpendicularly on the surface of the intestinal wall. The intestinal wall thickness was 2–3 mm.

During surgery (resection of the sigmoid and descending colon with lymphadenectomy), measurements on the intestinal surface were performed at certain points (Figure 2), which were of interest for recording the $StO_2$ level. Based on the obtained $StO_2$ values at these points, it was possible to judge the nature of the blood supply to the intestine, including in the position of the anastomosis. The adequate blood supply in the area of the anastomosis is the main guarantor of its consistency.

The analysis of hemoglobin oxygen $StO_2$ was performed at several positions along the colon relative to the tumor. The starting point was the center of the tumor. Measurements were carried out from the tumor borders with an approximate step of 5 cm, which was determined visually. At these points, measurements were carried out in three adjacent positions at the same level to increase the degree of measurement reliability, the distance between which was 1–2 mm.

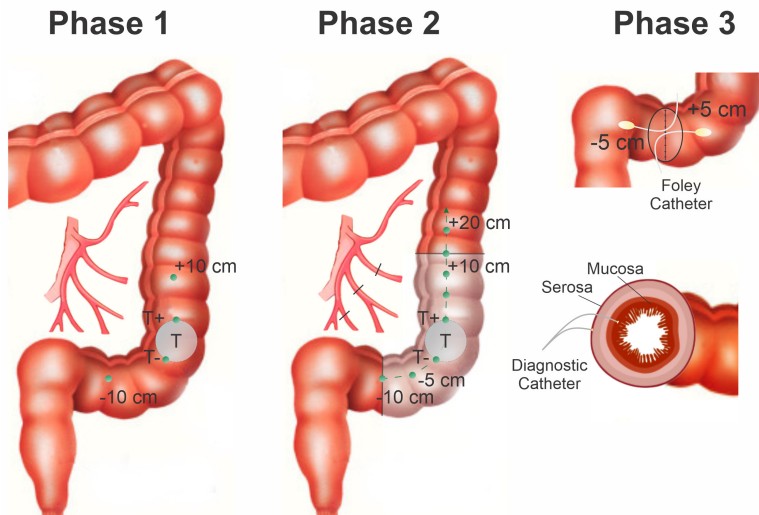

**Figure 2.** Scheme of the surgery with marked points for measuring the tissue saturation. Points 0 cm and 20 cm were the boundaries of the anastomosis. Tissue saturation of the tumor was measured at three points, where T− and T+ tumor borders, and T is the center of the tumor. The section of the intestine to be reshaped is highlighted in white. Black borders are the border of the anastomosis. At points +5 cm and −5 cm, the spectra of diffuse reflectance, transmission, and the degree of $StO_2$ were measured when a Foley catheter with a cylindrical diffuser was inserted into the intestine.

The measurements during the operation were divided into three stages. Stage I: measurement of $StO_2$ in the intestinal wall at the beginning of the operation before its mobilization. Measurements were carried out from the starting point (tumor invasion onto the outer intestinal wall) to the visible proximal and distal tumor boundaries (T+ and T− points). Further measurements were taken from the tumor borders up and down the intestine. Stage II: mobilization of a section of the intestine with a tumor and intersection of arterial and venous vessels was carried out. The obtained values of the $StO_2$ level at this stage make it possible to assess the blood supply to the intestinal area after mobilization, which is extremely important for establishing a reliable anastomosis. Stage III: resection of the sigmoid and descending colon was performed, and sutures were applied to the posterior part of the anastomosis. During the formation of the anastomosis, the $StO_2$ level was measured with the intraluminal position of the fiber optic diffuser, which was connected to a white light source. To enter the diffuser into the intestine, a Foley catheter filled with a scattering liquid (fat emulsion solution) was used to ensure the conditions for uniform propagation of light from the surface of the balloon and the equidistant position of the cylindrical diffuser from the intestinal walls. During the formation of the anastomosis along the resection line, $StO_2$ measurements were taken at two levels—serous-muscular and mucous-submucosal layers. During the operation, the spectra from the areas of the intestine with low $StO_2$ were measured; due to the disturbed circulatory bed of the intestinal tissues, some biological tissue acquired a visually blue tint.

### 3. Results and Discussion

#### 3.1. Mathematical Modeling of Diffuse Scattering and Transmission Spectra

Modeling of diffuse light scattering spectra in the intestinal wall was carried out, minimizing the difference between the simulated and experimental spectra by the least = squares method. Based on the data obtained, a graphical dependence of the attenuation coefficient on the wavelength is plotted (Figure 3). For the calculation, we used the linear, quadratic, cubic exponential functions of the scattering coefficient and a superposition of the exponential functions of Mie scatterers and Rayleigh scatterers. The coefficient characterizing the dependence of scattering on the wavelength for Mie scatterers was set as $a_1 = 1.5$, and for Rayleigh scatterers, $a_1 = 4$. In the simulation, the signal in the wavelength range of 520–590 nm was estimated.

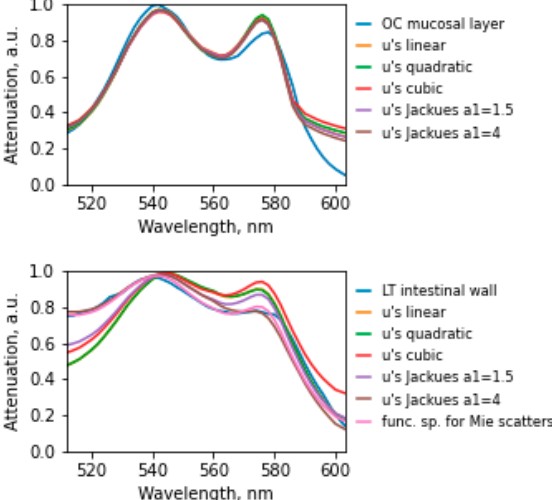

**Figure 3.** Experimentally obtained and corresponding simulated diffuse scattering spectra of the intestinal wall. The experimental spectral dependence of the attenuation coefficient on the wavelength for placing the fibers on one side of the wall (graph above) and "light through the wall" (graph below) is highlighted in blue.

The minimum discrepancy between the experimental spectrum (Table 1) and the model in placing the fibers on one side corresponded to the model with a linear dependence (Error = 0.0034) for the scattering coefficient and the empirical model with the coefficient $a_1 = 4$ (Error = 0.0029). The minimum error in the case of placing the fibers in the transmission was observed in the empirical model and amounted to Error = 0.0003 for the superposition of functions for Mie scatterers.

**Table 1.** The magnitude of the discrepancy (error) between the model and experimental spectra.

|  | One Side (OS) | Light Through (LT) |
|---|---|---|
| $\mu's$ linear | 0.0034 | 0.7449 |
| $\mu's$ quadratic | 0.0039 | 0.7452 |
| $\mu's$ cubic | 0.0054 | 0.1211 |
| $\mu's\ a_1 = 1.5$ | 0.0033 | 0.1906 |
| $\mu's\ a_1 = 4$ | 0.0029 | 0.0204 |
| func. superpos. for Mie scatters | 2.7220 | 0.0003 |

*3.2. Study of the Level of Hemoglobin Oxygen Saturation in the Intestinal Wall*

For each stage of the surgical operation, the $StO_2$ was obtained. Before the beginning of mobilization of the intestine in the center and at the border of the tumor, the level of oxygenation was 62.8%, and in the area of healthy tissue, the $StO_2$ level was 75.4%. Using the data obtained, it is possible to assess the nature of the blood supply to the intestine at different stages of the surgical intervention. These values do not correspond to the usual values that are recorded by a pulse oximeter since the method of optical spectroscopy records tissue saturation as it was explained earlier. $StO_2$ is most often understood as the degree of hemoglobin oxygenation in muscle tissue, in which the ratio of arterial and venous blood is 25/75%. According to several authors, the upper limit of $StO_2$ ranges between 81% and 86% [19]. Reduced values are considered to be $StO_2$ less than 70% [20,21].

The clinical part of the study involved 10 patients. Not all patients were able to obtain data at each stage of the operation due to various factors (limited visualization of part of the intestine at the stage, insufficient operation time, determination of the measurement method): six patients were measured for $StO_2$ before and after intestinal mobilization, eight patients were measured before mobilization and after anastomosis. Figure 4 shows a graphical dependence of the $StO_2$ level measurement before and after mobilization.

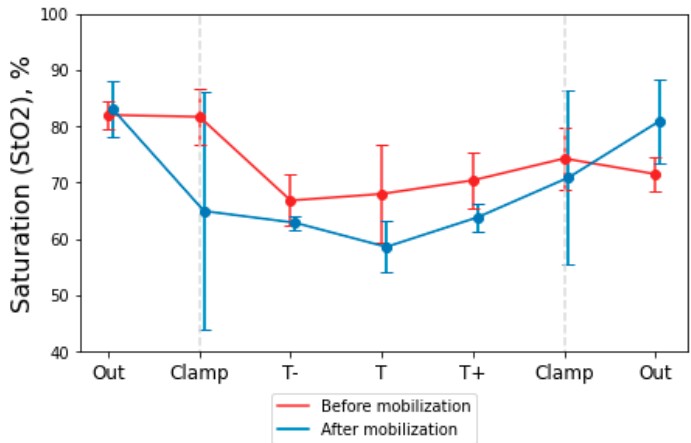

**Figure 4.** Graphical dependence of $StO_2$ from the measurement point before and after mobilization of the intestinal area. The dotted lines indicate the points where the surgical clamps were located. Points T− and T + are the borders of the tumor up and down the intestine; T is the center of the tumor.

Ligation and crossing of the great vessels were provided for by the scope of the surgical intervention. At the edges of the mobilized and reshaped intestine, the $StO_2$ level decreased to 55%, and at the border of the intestinal anastomosis, the $StO_2$ level was 82%.

Next, we investigated the dynamics of changes in intestinal $StO_2$ before mobilization and after anastomosis when parts of the intestine were sutured. According to the obtained data, the $StO_2$ changes in the intestine before mobilization and in the edges of the intestine after the anastomosis was determined (Figure 5). The post-anastomosis tissue saturation value ($StO_2$ = 76.4%) at the studied points in all patients compared to the pre-mobilization value ($StO_2$ = 73.2%). The postoperative period in all patients passed without complications, in particular, without inconsistency of the sutures of the anastomoses. All patients were followed up for at least 10 days after the operation until discharge from the hospital.

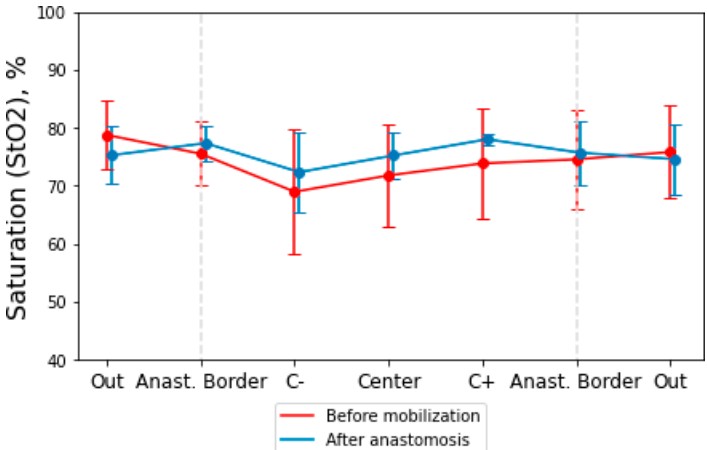

**Figure 5.** Dependence of intestinal $StO_2$ before mobilization and after anastomosis. The dotted lines mark the boundaries of the anastomosis. "Center" point is the center of zone of mobilization in the "Before mobilization" stage. In the "After anastomosis" stage, this point is the anastomosis suture position. "C+" and "C−" points are positioned higher and lower on the bowel from the center of the mobilization zone.

Below, the data of one clinical case (patient P.) are presented, which we have chosen for detailed consideration. Figure 6 shows the spectra of the light attenuation coefficient and histograms with the $StO_2$ values at the studied points before and after the mobilization of the intestinal area. The spectra obtained show characteristic oxyhemoglobin peaks at 542 nm and 576 nm. The uncertainty for each measurement is based on the standard deviation.

The deviation of the $StO_2$ values from the average at some points during the placement of the fibers from the side of the serous membrane of the intestine is due to technical problems of installing the light guide. They are associated with intestinal peristalsis and the patient's respiratory movements; therefore, it is difficult for the surgeon to maintain the required distance between the light guide and the intestinal wall. These factors lead to distortions of the transmission spectrum of light when receiving a signal.

During the mobilization of the intestine, the arterial and venous vessels feeding the intestine were ligated and transected. We compared the results at similar points in the intestine before the vessels ligation and after the ligation; the values before the vessels ligation were lower than after. The $StO_2$ decrease was due to a sharp decrease in the amount of oxyhemoglobin entering the tissues and a simultaneous increase in the concentration of reduced hemoglobin in the intestinal wall tissues over time. An increase in the concentration of reduced hemoglobin is associated with the diffusion of oxygen from oxyhemoglobin to the adjacent tissues cells. The time elapsed from crossing the vessels to the beginning of the measurement corresponded to 20–30 min.

Measurements were performed at the edges of the mobilized and reshaped bowel, where the blood supply was changed, and behind the clamps, where the blood supply was

regular. In Figure 6, the "Clamp" points correspond to the measurement in the area of the surgical clamps at the intestine site. The $StO_2$ values at the edges of the mobilized intestine are lower than at the points behind the applied clamps.

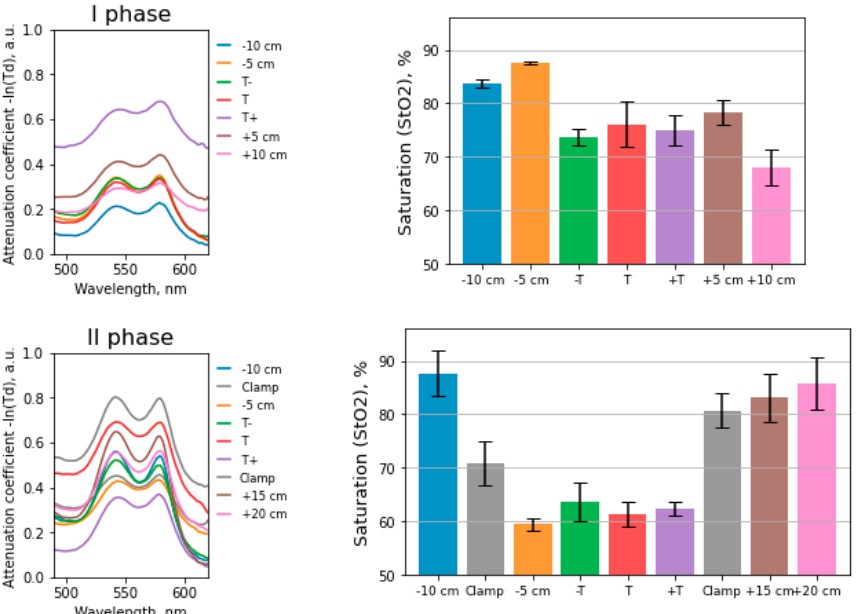

**Figure 6.** Absorption spectra at the points where tissue saturation was calculated. Histogram of tissue saturation values in the intestinal wall before and after mobilization.

Figure 7 shows the $StO_2$ values and light attenuation spectra when measured in layers and with an intraluminal diffuser at the 3rd stage of surgery. Measurements in transmitted light were carried out on the mobilized part of the intestine through an incision.

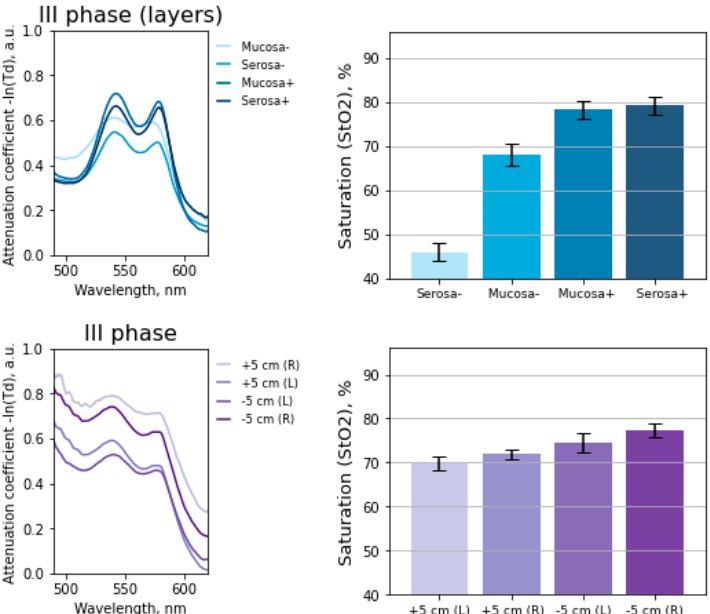

**Figure 7.** Histogram of tissue saturation values in the intestinal wall before anastomosis. The measurements were carried out in layers and with an intraluminal diffuser.

The root-mean-square deviation for measurements at each point with the intraluminal method ($\sigma = 1.6$) is lower than when the fibers are placed on one side ($\sigma = 2.1$). The deviation of the values may be associated with intestinal peristalsis, the patients respiratory

movements, and therefore the distance between the light guide and the intestinal wall changes over time.

In work [11], StO$_2$ was compared in anastomoses with and without leakage; a significant rise in the tissue saturation after the completion of the anastomosis in the anastomosis proximal part was seen in anastomoses without leakage. This rise was absent in anastomoses that ultimately developed leakage.

## 4. Conclusions

The most important problem in the treatment of colorectal cancer is preventing intestinal anastomosis leakage, which leads to the development of widespread peritonitis. One of the main predictors of this fatal complication is the technical aspects of surgery; the main of which is the inadequate blood supply to the anastomosed organs during the mobilization process. Unfortunately, the visual assessment of the blood circulation of the anastomosed organs in some cases is subjective and depends primarily on the experience of the operating surgeon. Even the most experienced of them can make tactical mistakes since visible changes from the side of the serous membrane occur later than from the side of the mucous membrane and are not visible visually at the time of surgery. However, these changes in blood supply can be discovered with the use of modern laser technologies, which make it possible to assess not only the vascular structure but also the level of oxygenation.

In this work, a method for assessing the risk of intestinal anastomosis leakage based on spectroscopic analysis of hemoglobin oxygen saturation and the level of blood supply of tissues is proposed and tested in a clinical setting, which makes it possible to objectify the assessment of the state of intestinal blood supply in the anastomosis area. A feature of the proposed method measures hemoglobin oxygen saturation in both geometries: diffuse reflectance and diffuse transmittance with illuminating fiber from the intestinal lumen while measuring the spectroscopic signal outside the intestinal wall. The present research has shown the relevance of the proposed method. In the future, this technique will make it possible to assess the level of tissue saturation of the anastomosis edges and thereby reduce the risk of developing anastomosis leakage and lethality of bowel surgeries.

**Author Contributions:** Conceptualization, V.B.L. and S.S.K.; Methodology, D.M.K., T.A.S., and V.B.L.; Formal analysis, D.M.K., T.A.S., V.B.L., V.V.L., and S.S.K.; Investigation, D.M.K., T.A.S., and T.A.M.; Resources, A.A.S., A.S.G., V.V.L., and S.S.K.; Writing—original draft preparation, review and editing, D.M.K., T.A.S., V.B.L., V.V.L., and S.S.K. All authors have read and agreed to the published version of the manuscript.

**Funding:** The research was carried out within the state assignment of fundamental scientific research for state academies of sciences of the GPI RAS (theme "Physical methods in medicine and biology" No. 0024-2019-0003).

**Institutional Review Board Statement:** The study was conducted according to the guidelines of the Declaration of Helsinki and approved by the Institutional Review Board of I.M. Sechenov First Moscow State Medical University.

**Informed Consent Statement:** Informed consent was obtained from all subjects involved in the study.

**Data Availability Statement:** The data supporting the findings of this study are available within the article.

**Conflicts of Interest:** The authors declare no conflict of interest.

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
