# Peer review of "Intraoperative Control of Hemoglobin Oxygen Saturation in the Intestinal Wall during Anastomosis Surgery"

_photonics, doi:10.3390/photonics8100427_

Round 1

Reviewer 1 Report

In the manuscript entitled “Intraoperative control of hemoglobin oxygen saturation in the intestinal wall during anastomosis surgery”, the authors use numerical and experimental methods to show a new approach for probing hemoglobin oxygen saturation in the intestinal wall during anastomosis surgery. The procedure followed during research is well detailed in the manuscript. Results are also well explained, and conclusions are in agreement with the main text. In general, I found the manuscript easy to read and follow. Therefore, I concluded that in terms of concepts, results and presentation, this manuscript has the level of novelty required to be published in this journal. However, I observed that there are several claims throughout the manuscript without any reference to previous literature. To give some examples:

  • in the first paragraph (lines 16 to 24) there are at least three different claims without any reference.
  • In paragraph 2 (lines 25 to 33) the authors only referenced one of their claims.

I would certainly recommend publishing this manuscript after the authors have addressed all these minor issues.

Author Response

We agree with comments and we added literature links to article. Added article pdf with highlight the corrections.

Reviewer 2 Report

The authors present spectrometer setup for intraoperative hemoglobin oxygen saturation assessment of the intestine wall. They propose two modes of measurement – reflectance and transmittance. The device is used during surgery on multiple patents and the oxygen saturation data before and after procedure in specific points along the intestine is shown. Clear differences in saturation can be detected. The paper is of relevance to the field biophotonics and medical professionals.

Here are my specific questions and suggestions:

  1. The figure 1. Should be rearranged, the zoom-ins on fiber ends should be enlarged, the fiber end which contacts the tissues is not indicated (ends abruptly in mid-air), it should at least be labeled ‘to sample’ or something like that. Usually such schemes should follow from the source to the detector/processor – here its 3,2,1 – reverse order.
  2. The figure 1 and 2 could be combined in one “mosaic” figure, just like in top journals.
  3. figure 3 – the labels and points are too small, T+ and T- is blurry.
  4. figure 6 – the label in the description in the text is “clamp” while in the figure it is “clip” – correct for naming consistency.
  5. The Monte Carlo model used to calculate <L> should depend on some specific (reduced) scattering coefficient of the medium – what is its value, or was it perhaps averaged somehow?
  6. Perhaps it should be added that equation 2 simply aims to minimize the chi-square of the sum of residuals of fitting the model data to the experimental data.
  7. The section 2.2. is not entirely clear. In the equation for light attenuation coefficient A, it seems to me that there is a mix-up between functions of wavelength (such as eHB(lambda)) and the coefficients c0,1,2 – shouldn’t they be also functions of the wavelength? I don’t mean that the algorithm is incorrect, but I think that the formula may be not formally precise. Also, the description of the whole algorithm is a bit vague and lacks details. Especially it is not entirely sure how the scattering part is calculated as a0 and a1 are not explained in equation 4.
  8. Figure 5 – the C- and C+ are not described.
  9. The section 3.2. seems out of place. After the experimental data is shown in the previous section then comes the discussion on modelling of diffuse scattering. It seems more feasible close to section 2.2. since this, if I understand correctly, is exactly the modelling required to establish the <L> as well as the model which is iterated to minimize fitting errors to the data, as in equation 2? Please move the section to proper place and improve the descriptions as now it’s a bit confusing.

Author Response

We agree with comments. 1) We enlarged the fiber end and marked the distal end. The order corresponds to the direction from source to detector. 2) Done. 3) Done. 4) Done. 5) Yes it is. Added to article. 6) Yes, it’s done. 7) The c_i are fitting coefficients that just describe the dependence of the scattering coefficient on the wavelength. That is, c_0 is the intercept and c_1 is the slope of the line, if we consider the linear regression option. For higher orders of the approximating polynomial, these are the coefficients in front of the corresponding expansion powers. 8) Done, added to article. 9) Thanks for the constructive comment. Replaced. 

Added article with highlighted corrections.
